# Impact of Foliar Application of Amino Acids on Essential Oil Content, Odor Profile, and Flavonoid Content of Different Mint Varieties in Field Conditions

**DOI:** 10.3390/plants11212938

**Published:** 2022-11-01

**Authors:** Aloyzas Velička, Živilė Tarasevičienė, Ewelina Hallmann, Anna Kieltyka-Dadasiewicz

**Affiliations:** 1Department of Plants Biology and Food Science, Faculty of Agronomy, Agriculture Academy Vytautas Magnus University, Donelaicio STR. 52, LT-44248 Kaunas, Lithuania; 2Department of Functional and Organic Food, Institute of Human Nutrition Sciences, Warsaw University of Life Sciences, Nowoursynowska 159 c, 02-776 Warsaw, Poland; 3Bioeconomy Research Institute, Agriculture Academy, Vytautas Magnus University, K. Donelaicio Str. 58, LT-44248 Kaunas, Lithuania; 4Department of Plant Production Technology and Commodity, University of Life Sciences in Lublin, Akademicka 15, 20-950 Lublin, Poland

**Keywords:** aromatic amino acids, *Mentha spicata*, *Mentha piperita*, secondary metabolites

## Abstract

Mint is an industrial plant that is a good source of essential oil and many phenolic compounds that have several positive benefits to human health and can be used to prevent the development of many diseases. The aim of this research was to investigate the possibility of increasing essential oil and flavonoid content, changing the chemical composition of these compounds in different mint cultivars under foliar application with precursors (phenylalanine, tryptophan, and tyrosine) at two concentrations, 100 and 200 mg L^−1^, to enable the possibilities for wider use of these plants when they are grown in field conditions. Spraying with phenylalanine at 100 mg L^−1^ concentration increased essential oil content in *Mentha piperita* ‘Granada’ plants by 0.53 percentage units. Foliar application with tyrosine solutions at 100 mg L^−1^ concentration most effectively influenced the essential oil odor profile *Mentha spicata* ‘Crispa’. The highest number of total flavonoids was in *Mentha piperita* ‘Swiss’ sprayed with tyrosine at 100 mg L^−1^ concentration. The flavonoid content depended on the mint cultivar, amino acids, and their concentration. The results showed that the effect of amino acid solutions on different secondary metabolites’ quantitative and qualitative composition differed depending on the mint cultivar; therefore, amino acids and their concentrations must be selected based on the cultivar they are targeting.

## 1. Introduction

*Mentha* belongs to the *Lamiacea* family, known as mint, including 19 mint species and 13 hybrids [1]. According to Telci et al. [2], of all the mint species, *Mentha canadensis*, *Mentha piperita*, and *Mentha spicata* are the major mint species with the most economic importance due to their medicinal and aromatic values. The essential oil from citrus and mint are the most important in the world‘s essential oil trade [3]. Mint essential oil and extracts are used in perfumery, confectionery, and for pharmaceutical preparations. Medicinal plant applications in food and pharmaceutical industries depend on the amount of biologically active substances and the chemical composition that they contain. According to Verma et al. [4], essential oil content in *M. piperita* ranged from 0.19 to 0.60% depending on the mint variety and age of the plant. According to Korousou et al. [5], in *M. spicata* mint grown in Mediterranean countries, essential oil content ranged from 0.19 to 1.2%. The main groups of essential oil in mint are alcohols, ketones, esters, ethers, and oxides [2]. The main compounds of mint essential oil belonging to the *Mentha* L. genus are menthol, menthone, isomenthone, menthyl acetate, linalool, linalyl acetate, pulegone, carvone, and piperitenone oxide [2]. Plants accumulate secondary metabolites in small amounts. Scientists state that these biologically active substances consist of only 1% of the plant’s dry weight and the number of compounds depends on the plant species and stage of development [6]. Biosynthesis and accumulation of secondary metabolites depend on genetic and environmental factors (light, temperature, soil water, soil fertility, and salinity) [7]. Plants’ secondary metabolites are the molecules which play an important role in the plant’s interaction with the environment as it adapts to both biotic and abiotic stress conditions [8]. Despite the high amount of essential oil, mint plants are rich in phenolic compounds [9]. These include flavonoids such as luteolin, apigenin, acacetin, diosmin, salvigenin, and thymonin, and flavonols such as catechin, epicatechin, and coumarins; aesculetin and scopoletin compounds account for 10–70% of the total amount of mint phenols [10,11]. According to Benabdallah et al. [12], total flavonoid content in mint grown in Algeria fluctuated from 9.90 to 31.77 (mg of rutin equivalent per g of dry weight), depending on the species of mint. Researches show that essential oil and extracts from mint have significant effects such as analgesic, anti-inflammatory, antipyretic, DNA damage protecting activity, antioxidant, antiandrogenic, antimicrobial, cytotoxic, antiviral, anticancer, antiemetic, antibacterial, antiallergic, antiparasitic, sedative, antichlamydial, anticholinesterase, hepatoprotective, antispasmodic, acute toxicity effect, antimutagenic, and cardiovascular [13,14]. In practice, for these compounds, biostimulants composed of mineral elements can be used for stimulation; these include humic substances, vitamins, amino acids, chitin, chitosan, poly, and oligosaccharides [15]. Several studies have shown that foliar spray with amino acids or biostimulants based on amino acids can increase yield and biologically active substances’ content as well their chemical composition in *Satureja hortensis* L., *Calendula officinalis* L., *Mentha piperita* var. citrata., *Urtica pilulifera* L., *Ocimum basilicum* L., *Nigella sativa* L., *Achillea millefolium* L., *Coriandrum sativum* L., and *Foeniculum vulgare* Mill plants [16,17,18,19,20,21,22,23,24]. According to Pratelli and Pilot [25], amino acids play significant roles in catalyzing the reactions of the secondary metabolism in plants. Aromatic amino acids (phenylalanine, tryptophan, and tyrosine) have a significant role in the shikimate pathway because they are derived from the precursor chorismate, which is the final product of this pathway [26]. According to Tegeder and Masclaux-Daubresse [27], depending on plant species and environmental conditions, plants reduce inorganic nitrogen to amino acids in nodules, roots, and leaves. The transport of amino acids is generally related to the transport of proteins in plants. Amino acids in plants are transported by different transporters, for example: AAP (amino acid permeases), LHT (lysine and histidine transporters), ProT (proline transporters), GAT (γ-aminobutyric acid transporters), ANT (aromatic and neutral amino acid transporters), AUX (auxin transporters), ATL (amino acid transporter-like), and VAAT (vesicular aminergic-associated transporters) [28]. Phenylalanine, tyrosine, and tryptophan, in addition to their role as constituents of proteins, are the precursors of many secondary metabolites. For example, tryptophan is the precursor of auxin, alkaloids, phytoalexins, and indole glucosinolates, while tyrosine is the precursor of chinons, betalains, and isoquinoline alkaloids; coumarate is a precursor to flavonoids, coumarins, lignins, lignans, phenolic esters, and synthesis of phenolic waxes [29,30]. According to Croteau et al. [31], the most important of these amino acids is phenylalanine, from which more than 8000 compounds are derived. The most abundant of these compounds is lignin; among the others are phenolic acids, anthocyanins, flavonoids, isoflavonoids, tannins, volatiles, and phenylalanine, which is important for the synthesis of gibberellins [30,31]. Research has shown that foliar spray at different concentrations with phenylalanine and tyrosine solutions had a significant positive effect on the total amount of phenols, essential oil, and their composition in *Coleus blumei* L., *Melissa officinalis* L., and *Ocimum basilicum* L. plants [32,33,34]. Noviyanti et al. [35] determined that use of sucrose, phenylalanine, and tyrosine solutions at high concentrations led to the best results for kaempferol (733.33, 911.11, and 869.44 mg L^−1^/g dry weight) and quercetin (165.00, 245.83, and 233.33 mg L^−1^/g dry weight) content in adventitious root culture of *Gynura procumbens*. Sachet et al. [36] found that foliar application with phenylalanine at 150 mg L^−1^ concentration of *Anethum graveolens* L. increased herb oil yield by 0.144–0.256 g and flavonoid content by 0.163–0.262 g compared with unsprayed plants. Poorghadir et al. [37] determined that foliar application with proline and phenylalanine at a concentration of 1.5 g L^−1^ was the most effective for essential oil, carvacrol, and gama terpinene content in *Satureja hortensis* plants; compared with unsprayed plants, amounts of these compounds increased several times. Samani et al. [38] determined that foliar application with phenylalanine at 250 and 500 mg L^−1^ concentration increased—compared with unsprayed plants—the amount of essential oil in *Salvia officinalis* plants by 0.06–0.12 (mL 100 g^−1^ DW) and the amount of 1,8-Cineole (0.97% DW) and camphor (0.64% DW). Aghaei et al. [39] states that phenylalanine at 500 mg L^−1^ concentration, in combination with arbuscular mycorrhizal fungi, vermicompost, and manure, increased essential oil content in *Hyssopus officinalis* plants by 40% after one year, and 80% after two years, in comparison to the control. Flavonoids and terpenoids have important ecological functions in plants. They protect plants against various biotic and abiotic factors and serve as attractants. Their odor, color, and taste attract pollinators and seed-dispersing animals [40,41]. Other research results show that plants’ secondary metabolites are important for plant growth and development, and that a higher content of these compounds demonstrates a plant‘s possibilities for use in human consumption. The aim of this work was to determine the impact of foliar application of amino acids on the biologically active compound content of different mint varieties in field conditions. The hypothesis of this work was: presumably not only do the content and composition of flavonoids and the odor profile of the essential oil depend on the mint species, cultivar, and the abiotic environment, but they may also be optimized during the vegetation by using aromatic amino acids.

## 2. Results

The total essential oil content in the mint fluctuated from 1.19 in *M. spicata* ‘Crispa’ to 3.24% in *M. piperita* ‘Swiss’ (Table 1). The positive effect of amino acids was only discovered when mint of the *M. piperita* ‘Granada’ species was sprayed with phenylalanine solution of 100 mg L^−1^ concentration. Compared with unsprayed plants, the amount of essential oil in mint sprayed with this solution increased by 0.53 percentage units. Compared with unsprayed mint, mint of the *M. spicata* ‘Crispa’ species sprayed with amino acids had a smaller amount of essential oil, except for that sprayed with the phenylalanine solution of 100 mg L^−1^ concentration, which, compared with unsprayed mint, had no significant effect on the amount of essential oil. The amount of essential oil found in mint of the *M. piperita* ‘Swiss’ species sprayed with water was 1.19 percentage units smaller than in the unsprayed mint (Table 1). Assessing the impact of amino acids on the amount of essential oil in mint of the *M. piperita* ‘Multimentha’ species, no effect was noticed. The significant effect on the accumulation of essential oil in mint leaves was noticed when examining traits of particular species. Compared with unsprayed mint, the significantly largest amount of the component with this chemical composition was determined in *M. piperita* ‘Swiss’ mint, whereas the fundamentally smallest amount of the component was noticed in mint of the *M. spicata* ‘Crispa’ and *M. piperita* ‘Granada’ species (Table 1).

A principal component analysis (PCA) was performed to evaluate the relationships between the applications of the amino acids and mint’s essential oil odor profile during the period 2017–2019. The first factor (PC1) explained 45.85%, while the second factor (PC2) explained 33.63% of the total variance (Figure 1). As can be seen in Figure 1, the separation of the samples clearly occurs based on mint varieties and foliar application of amino acids. For the *M. piperita* ‘Multimentha’ (marked in black color) and *M. piperita* ‘Swiss’ (marked in purple color) varieties, the essential oil aroma profile of unsprayed mint and mint sprayed with amino acids obtained positive PC1 values. For the *M. piperita* ‘Granada’ variety, the aroma profile of unsprayed mint and mint sprayed with amino acids obtained positive PC2 values. For the *M. spicata* ‘Moroccan’ (marked in green color) and *M. spicata* ‘Crispa’ (marked in blue color) varieties, the essential oil odor profile of unsprayed mint and mint sprayed with amino acids obtained negative PC1 values. PCA analysis clearly shows that mint’s essential oil aroma profile and the effect of solutions of amino acids depended on mint varieties. However, foliar application with tyrosine solution of 100 mg L^−1^ was the most effective in influencing the essential oil odor profile of *M. spicata* ‘Crispa’ and *M. spicata* ‘Moroccan’ (Figure 1). 

It was determined that mint’s essential oil odor profile depended on the mint’s species and variety, and that the essential oil profile of *M. spicata* is different compared with *M. piperita*. A total of 48 volatile compounds belonging to different groups of chemical compounds have been preliminarily identified in mint essential oil, including aldehydes, ketones, esters, alcohols, sulfides, terpenes, and heterocyclic compounds. Analysis of results has shown that 23.53% of all identified compounds were terpenes, 19.61% were ketones, and 19.61% were esters (Appendix A). The main compounds of terpenes were: alpha pinene, alpha phellandrene, beta-phellandrene, limonene, terpinolene, *p*-cymene, limonene oxide, geraniol, citronellol and thymol, while the main compounds of ketones were: propanone, acetoin, 2-acetyl-1-pyrroline, cumene, beta demascone, alpha ionone, 2-tridecanone, beta ionone, and acetophenone, and the main compounds of esters were: ethyl acetate, isopropyl acetate, butyl acetate, ethyl butyrate, butanoic acid, isoamyl acetate, methyl butanoate, eugenol, and ethyl hexanote. In *M. spicata* ‘Moroccan’ sprayed with solutions of phenylalanine and tyrosine at both concentrations and tryptophan of 100 mg L^−1^ concentration, the compound sabinene was identified, while in *M. spicata* ‘Crispa’ sprayed with tyrosine solution at a concentration of 200 mg L^−1^, phellandrene was found. Limonene was identified in *M. spicata* ‘Moroccan’, *M. piperita* ‘Granada’, and *M. piperita* ‘Multimentha’ essential oil after the application of phenylalanine of 200 mg L^−1^ concentration. Limonene oxide was preliminarily identified in *M. spicata* ‘Crispa’ following the application of tryptophan at 200 mg L^−1^ concentration and of tyrosine at 100 mg L^−1^, and was identified in *M. piperita* ‘Multimentha’ following the application of tyrosine at a concentration of 100 mg L^−1^. Citronellol was preliminarily identified in *M. spicata* ‘Moroccan’ following the application of tyrosine at 100 mg L^−1^ concentration and in *M. spicata* ‘Crispa’ following the application of phenylalanine of 200 mg L^−1^ concentration (Appendix A). 

It was observed that the amount of flavonoids was influenced by mint variety as well as foliar application of amino acids (Figure 2). The total amount of flavonoids in the mint fluctuated from 35.38 in *M. spicata* ‘Crispa’ to 221.15 mg 100 g^−1^ in *M. piperita* ‘Swiss’. Compared with unsprayed plants, foliar application with water increased total flavonoid content in *M. spicata* ‘Moroccan’, *M. spicata* ‘Crispa’, and *M. piperita* ‘Granada’ by 1.27, 1.22, and 1.55 times, respectively. Different effects of water application compared with unsprayed plants was determined in *M. piperita* ‘Swiss’ plants: the amount of total flavonoids decreased 2.67 times. The effect of phenylalanine in the mint was different, but this amino acid solution was the most effective at a concentration of 200 mg L^−1^. Foliar application with this amino acid solution increased the amount of total flavonoids in *M. spicata* ‘Moroccan’, *M. spicata* ‘Crispa’, and *M. piperita* ‘Multimentha’ mint, compared with unsprayed mint by 1.98, 1.41, and 3.94 times, respectively, while in *M. piperita* ‘Swiss’, spraying with a concentration of 200 mg L^−1^ decreased total flavonoids 1.45 times. Results showed that foliar application with amino acids, except for tyrosine, of 100 mg L^−1^ concentration negatively influenced total flavonoid content in *M. piperita* ‘Swiss’ mint compared with unsprayed plants. The effect of tryptophan on total flavonoid content depended on the mint cultivar. For example, in *M. spicata* ‘Moroccan’ plants, the flavonoid content, compared with unsprayed mint, decreased, while it increased in *M. spicata* ‘Crispa’ and in *M. piperita* ‘Multimentha’. In assessing the effects of tyrosine on total flavonoid content in mint, it was determined that foliar application at a concentration of 100 mg L^−1^ produced the highest amount of these compounds—compared with unsprayed plants—in *M. spicata* ‘Crispa’ (1.90 times more) and *M. piperita* ‘Granada’ (1.45 times more), while application at a 200 mg L^−1^ concentration increased total flavonoid content by 1.33 times in *M. spicata* ‘Moroccan’ mint (Figure 2).

The water application was the most effective for rutin content in *M. spicata* ‘Moroccan’ and in *M. piperita* ‘Swiss’ plants; the amount of this compound, compared with unsprayed plants, increased by 3.04 and 6.60 times (Table 2). Foliar application with amino acids produced the highest rutin content in *M. piperita* ‘Swiss’ plants, and the highest amount of this compound was identified in plants that were sprayed with water and tyrosine at a concentration of 100 mg L^−1^. In addition, application with this concentration increased rutin content by 1.46 times, compared with unsprayed plants, in *M. piperita* ‘Granada’. Foliar-applied solutions influenced the amount of rutin differently. For example, compared with unsprayed plants, foliar application with phenylalanine of 100 and 200 mg L^−1^ concentration negatively affected the amount of rutin in *M. piperita* ‘Granada’ mint, while in *M. piperita* ‘Multimentha’, the amount of this compound decreased—compared with unsprayed plants—when the plants were treated with tryptophan and tyrosine of 100 mg L^−1^ concentration solutions. Similarly, the amount of this compound decreased in *M. spicata* ‘Moroccan’, compared with unsprayed plants, when plants were treated with tryptophan of 100 and 200 mg L^−1^ concentration (Table 2). 

Isoquercetin content in the mint fluctuated from 2.33 in *M. piperita* ‘Multimentha’ to 7.30 mg 100 g^−1^ in *M. piperita* ‘Swiss’. Application with water produced the highest content of isoquercetin in *M. piperita* ‘Swiss’ and in *M. piperita* ‘Multimentha’ cultivars compared with unsprayed plants (Table 3). Application with phenylalanine of 100 mg L^−1^ concentration negatively affected this compound’s content, compared with unsprayed plants, by 1.44 times in *M. piperita* ‘Multimentha’, while in *M. piperita* ‘Swiss’, isoquercetin content decreased with the application of phenylalanine of 200 mg L^−1^ concentration. Phenylalanine and tryptophan of 200 mg L^−1^ concentration increased this compound content by 1.16 and 1.32 times, respectively, in *M. spicata* ‘Crispa’ cultivar plants. Application with tryptophan concentrations of 100 and 200 mg L^−1^ negatively affected isoquercetin content in *M piperita* ‘Granada’ mint compared with unsprayed plants. It was determined that *M. spicata* ‘Crispa’, *M. piperita* ‘Swiss’, and *M. piperita* ‘Multimentha’ cultivars accumulate isoquercetin in larger quantities (Table 3).

The results showed that application with amino acids had no negative influence on myricetin content in all mint cultivars. Compared with unsprayed plants, foliar application of amino acids and water solution increased myricetin content by 1.40–2.17 times in *M. piperita* ‘Swiss’ plants, while in *M. piperita* ‘Multimentha’ there was no significant increase of this compound, compared with unsprayed plants (Table 4). In the *M. piperita* ‘Granada’ mint cultivar, application with phenylalanine of 100 mg L^−1^ concentration and tyrosine of 200 mg L^−1^ concentration produced 2.94 and 1.94 times higher content of myricetin, compared with unsprayed plants. Foliar application with tyrosine at 100 mg L^−1^ concentration was the most effective on the myricetin content in *M. spicata* ‘Crispa’ plants. In assessing the influence of variety, it was determined that *M. piperita* ‘Granada’ mint is characterized by higher levels of this compound (Table 4). 

The quercetin content in mint fluctuated from 2.76 mg 100 g^−1^ in *M. piperita* ‘Granada’ to 115.91 mg 100 g^−1^ in *M. spicata* ‘Moroccan’. The results showed that foliar application with water can increase the content of this compound in mint (Table 5). For example, the quercetin content in *M. spicata* ‘Moroccan’ increased 1.30 times; in *M. piperita* ‘Granada’, it increased 5.89 times; and in *M. piperita* ‘Swiss’ mint cultivars, it increased 1.96 times compared with unsprayed plants. The positive effect on quercetin content of phenylalanine depended on mint cultivar and solution concentration. The highest amount of quercetin in *M. piperita* ‘Swiss’ was identified when plants had tryptophan of 200 mg L^−1^ concentration applied, while in *M. piperita* ‘Multimentha’, the highest amount was identified when tryptophan of 100 mg L^−1^ concentration was applied. The influence of tryptophan solutions in *M. spicata* ‘Moroccan’ mint was different: the amount of quercetin decreased 2.22 and 4.91 times compared with unsprayed plants. In assessing tyrosine’s effect on quercetin content in mint, it was determined that 100 mg L^−1^ concentration of this amino acid increased this compound’s content by 12.61 times in *M. spicata* ‘Crispa’ plants. However, the tyrosine solutions of higher concentration decreased this compound’s content in *M. spicata* ‘Crispa’, *M. spicata* ‘Moroccan’, and *M. piperita* ‘Swiss’. Compared with unsprayed mint, it was determined that *M. piperita* ‘Swiss’ mint accumulated 16.50 times more quercetin than *M. piperita* ‘Granada’ plants (Table 5).

The amount of apigenin in water-treated *M. spicata* ‘Moroccan’, *M. spicata* ‘Crispa’, and *M. piperita* ‘Multimentha’ mint was similar to that in unsprayed plants. The application with water in *M. piperita* ‘Granada’ plants, compared with unsprayed plants, decreased the amount of apigenin by 1.77 times, while in *M. piperita* ‘Swiss’, the effect was different: the amount of this compound increased by 2.11 times (Table 6). The highest amount of apigenin was determined in *M. spicata* ‘Moroccan’ plants treated with tryptophan of 100 mg L^−1^ concentration solution; however, the amount of this compound, compared with unsprayed plants, increased slightly. The application of phenylalanine of 200 mg L^−1^ solution in this mint cultivar decreased the amount of apigenin by 2.11 times. The effect of this amino acid at the same concentration was different in *M. spicata* ‘Crispa’, where the amount of this flavone increased by 1.77 times, while application with tyrosine decreased the amount of apigenin by 1.72–3.56 times. Tyrosine of 200 mg L^−1^ solution negatively affected the amount of apigenin in *M. piperita* ‘Granada’ mint, compared with unsprayed plants and those sprayed with other amino acid solutions. The highest amount of apigenin in *M. piperita* ‘Swiss’ was identified when plants were treated with water, tryptophan, and tyrosine solutions at 100 mg L^−1^ concentration, and the amount of this compound, compared with unsprayed plants, increased by 2.11, 2.51, and the 2.80 times, respectively. The application with tryptophan of 100 mg L^−1^ concentration solution significantly increased the amount of apigenin in *M. piperita* ‘Multimentha’ plants (by 1.41 times compared with unsprayed plants), while application with phenylalanine of 100 mg L^−1^ and tryptophan of 200 mg L^−1^ concentrations produced a different effect. Irrespective of the foliar application, the highest amount of apigenin was in *M. spicata* ‘Moroccan’ mint (Table 6).

The kaempferol content in mint fluctuated from 4.45 in *M. piperita* ‘Multimentha’ to 62.50 mg 100 g^−1^ in *M. spicata* ‘Moroccan’. Foliar application with tyrosine at 200 mg L^−1^ concentration was the most effective in *M. spicata* ‘Moroccan’ plants; the amount of this compound, compared to unsprayed plants, increased by 4.40 times (Table 7). Foliar application with water and other amino acid solutions made no significant difference to the amount of kaempferol in *M. spicata* ‘Moroccan’ mint. The highest amount of this compound in *M. spicata* ‘Crispa’ was identified—compared with unsprayed plants—when plants were sprayed with tryptophan solution at 100 mg L^−1^ concentration, while in *M. piperita* ‘Granada’ this amino acid solution had the least effect, compared with other solutions. In *M. piperita* ‘Swiss’ mint, the most effective for kaempferol accumulation was tyrosine at 200 mg L^−1^ concentration; compared with unsprayed plants, this produced an increase of this compound by 2.19 times. Phenylalanine solutions increased the amount of kaempferol in *M. piperita* ‘Multimentha’ by 2.55–11.04 times depending on the concentration, while application with tryptophan of 100 mg L^−1^ concentration increase the amount of this compound—compared with unsprayed plants—by 6.89 times. It was determined that, irrespective of foliar application, the highest amount of kaempferol accumulated in *M. spicata* ‘Moroccan’ and *M. piperita* ‘Granada’ mint (Table 7).

In order to assess the relations between spraying with solutions of amino acids and the total amount of flavonoids and their qualitative composition, a principal component analysis (PCA) was performed. The first two components (PCs) were associated with eigenvalues higher than one, and explained 42.80% and 20.54% of the total variance (Figure 3) The first factor (PC1) was highly and positively correlated with total flavonoids, quercetin, myricetin, isoquercetin, and rutin, whereas the second factor (PC2) was positively related to kaempferol and negatively related to apigenin. The principal component analysis revealed that the greatest total amount of flavonoids, quercetin, rutin, and isoquercetin were identified in *M. piperita* ‘Swiss’ mint, in both unsprayed plants and those sprayed with water and a solution of amino acids. Application to *M. spicata* ‘Moroccan’, *M. spicata* ‘Crispa’, and *M. piperita* ‘Granada’ was positively related to kaempferol. Spraying solutions of amino acids had a negative effect on the amount of apigenin found in almost all cultivars of mint (Figure 3). After hierarchical cluster analysis of mint sprayed with different solutions of amino acids, mint samples were grouped into three clusters (C1, C2, and C3) (Figure 4). The first cluster (C1) included 21 mint varieties sprayed with a solution of amino acids; these varieties contained the greatest amount of apigenin. The second cluster (C2) included nine mint varieties sprayed with solutions of amino acids; mint in this cluster contained the highest amount of rutin, isoquercetin, quercetin, and total flavonoids. The third cluster (C3) included 10 varieties sprayed with solutions of mint amino acids; these contained the greatest amount of myricetin and kaempferol. *M. spicata* ‘Moroccan’ mint sprayed with tryptophan of 200 mg L^−1^ concentration had the largest amount of apigenin, *M. piperita* ‘Swiss’ sprayed with tyrosine of 100 mg L^−1^ concentration had the largest amount of rutin, isoquercetin, quercetin, and total flavonoids, while *M. piperita* ‘Granada’ sprayed with tryptophan of 200 mg L^−1^ concentration had the largest amount of kaempferol, while the same variety sprayed with phenylalanine of 100 mg L^−1^ concentration had the largest amount of myricetin (Figure 4).

## 3. Discussion

*Mentha* species are widely cultivated in the world as industrial crops for essential oil production and as a source of different phenolic compounds for extracts. Research has been performed to investigate the possibilities for stimulating the synthesis of these secondary metabolites by using aromatic amino acids in vivo, in vitro, and in field conditions with different plants; however, there remains inadequate data on how effective aromatic amino acids as precursors are for the promotion of the synthesis of terpenoids and flavonoids in mint in field conditions [42,43,44,45]. The experiment’s results showed that stability was not found throughout the years of the experiment, nor was stability found in the statistically significant interaction between the years and the amounts of essential oil, essential oil odor profile, and flavonoids. In addition, it was determined that the cultivars and treatment, as well the interaction of these treatments, were statistically significant (*p* < 0.05). Assessing the impact of the positive effect of amino acid solutions on the amount of essential oil was determined only in *M. piperita* ‘Granada’ mint sprayed with phenylalanine of 100 mg L^−1^ concentration; this amino acid increased essential oil by 1.38 times. Foliar application with tyrosine of 100 mg L^−1^ concentration had the greatest influence on the essential oil odor profile in *M. spicata* ‘Crispa’ and *M. spicata* ‘Morrocan’ mint. Our results agree with those of EL-Zefzafy et al. [42] and Reham et al. [34], who found aromatic amino acids had a positive effect on the amount of essential oil and in the chemical composition of *Artemisia abrotanum* and *Ocimum basilicum* plants. The highest essential oil content (4.45%), yield of essential oil (1.16 mL/plant), and chamazulen content (22.37% and 23.50%) in *Artemisia abrotanum* plants were obtained in plants treated with phenylalanine of 250 mg L^−1^ concentration, while foliar application with phenylalanine of 50 mg L^−1^ and 100 mg L^−1^ concentration increased essential oil content in *Ocimum basilicum* plants by 0.04–0.10 percentage units, compared with unsprayed plants. There are several opinions regarding the influence of amino acids on essential oil content in plants. Firstly, amino acids affect the activity of enzymes and the metabolism of essential oil [46]. Carbohydrates, fatty acids, and amino acids represent the natural carbon pools for flavor compounds, which can also be liberated from their polymers. By amino acid degradation, phenyl propane benzenoids can be formed. From these, alkohols, aldehydes and esters can be obtained during hydrolysis of cyanogenic glycosides, from which aldehydes and ketones can be synthesized [47]. Secondly, amino acids are a source of energy, carbon, and nitrogen, which constitute plant tissues and organs [48]. According to Jiao et al. [49], phenylalanine can be used as a source of nitrogen. This macro element is one of the most important compounds for plant growth and development; it affects many enzymes that control plant physiological processes, essential oil formation, and production, and is found in all plant cells, plant proteins, hormones, and chlorophyll [50]. Nitrogen influences the production of essential oil through carbon metabolism and acetyl-CoA formation via the mevalonate pathway [51]. Positive effects of different concentrations of nitrogen on essential oil content and chemical composition were observed in different plants (mint, parsley, black cumin, and coriander) [52,53,54,55,56]. It was revealed that, depending on the variety of mint, the treatment of all used amino acids produced an increase in the amount of the total number of flavonoids compared with the unsprayed plants. For example, phenylalanine of 200 mg L^−1^ concentration in *M. spicata* ‘Moroccan’ increased flavonoids by 1.98 times, in *M. spicata* ‘Crispa’ it by 1.41 times, and in *M. piperita* ‘Multimentha’ by 3.94 times; tryptophan at 100 mg L^−1^ concentration increased flavonoids by 1.27 times in *M. spicata* ‘Crispa’ and by 2.08 times in *M. piperita* ‘Multimetha’; tryptophan at 200 mg L^−1^ concentration increased flavonoids by 1.35 times in *M. spicata* ‘Crispa’; tyrosine at 100 mg L^−1^ concentrate increased flavonoids by 1.90 times in *M. spicata* ‘Crispa’ and by 1.45 times in *M. piperita* ‘Granada’; and tyrosine at 200 mg L^−1^ concentration increased flavonoids by 1.33 times in *M. spicata* ‘Moroccan’. These results may be related to the fact that phenylalanine and tyrosine are the precursors of cinnamic acids, esters, coumarins, phenylpropenes, chromones, stilbenes, antrachinones, chalcones, flavonoids, isoflavonoids, neoflavonoids and their dimers and trimers, lignans, and neolignans [57]. According to Harborne [58], phenolic compounds are synthesized via the shikimate-phenylpropanoid-flavonoid pathway. Phenylalanine ammonia-lyase (PAL) and tyrosine ammonia-lyase (TAL) are regulatory enzymes involved in the phenylpropanoid pathway [59,60]. According to Feduraev et al. [61], adding phenylalanine and tyrosine can improve PAL and TAL activity. Barros and Dixon [30] state that the first step in the phenylpropanoid pathway for flavonoid synthesis is the deamination of phenylalanine into cinnamate by phenylalanine ammonia-lyase; subsequent formation of coumarate (which is the precursor of flavonoids) involves direct hydroxylation of the aromatic ring by *trans*-cinnamate 4-hydroxylase. Some plant families can also form coumarate directly from tyrosine by bifunctional phenylalanine—tyrosine ammonia-lyase (PTAL). Feduraev et al. [61] state that tyrosine can be more effective for flavonoid synthesis because phenylalanine transformation to *p*-coumaric acid has two reactions, whereas tyrosine needs only one. The positive effect of phenylalanine and tyrosine on flavonoid synthesis in *Scutellaria baicalensis*, *Ocimum tenuiflorum*, and *M. piperita* plants in vitro was observed by Kawka et al. [44], Jacob and Thomas [43], and Roy et al. [45]. Demirci et al. [62] found a positive effect of 24-epibrassinolide and phenylalanine combination on total flavonoid synthesis in hairy root culture of *Echinacea purpurea* L. Moench. Tryptophan is a precursor for the synthesis of many secondary metabolites: phytoalexins, glucosinolates, alkaloids, and auxins [63]. Tryptophan is the precursor of auxin, which coordinates plants‘ tropisms and regulates numerous developmental responses, including control of cell division, growth, and differentiation. In addition, this hormone is important for plants’ stress control, directly and indirectly [64,65,66]. Furthermore, the biosynthesis of the aromatic amino acids is an example of a feedback mechanism, which means that a higher production of tryptophan will induce the carbon flux towards the production of phenylalanine and tyrosine [67]. Tryptophan influence on total flavonoid content in *Nasturtium officinale* microshoots was observed by Klimek-Szczykutowicz et al. [68]. However, the authors state that the most effective amino acid for improving total flavonoid synthesis was phenylalanine. Our results showed that, not only are aromatic amino acids important for flavonoid synthesis, but they are also important for their concentration. According to Ouyang et al. [69], an excess of precursors can inhibit feedback in the metabolic pathway so that the synthesis of the compounds is not stimulated, but instead is inhibited. Rutin (3,3′,4′,5,7-pentahydroxyflavone-3-rhamnoglucoside) is flavonol that has antibacterial, antifungal, antimycobacterial, larvicidal, antimalarial, antiretroviral, and organ-protecting effects [70]. Rutin is an activator that improves plant disease resistance to bacterial pathogens via the rutin priming defense signal, which is modulated by the salicylic acid-dependent pathway [71]. El-Ashry et al. [72] found that the maximum rutin content in *Gardenia jasminoides* Ellis calli cultures was produced with phenylalanine at 3 mg L^−1^ concentration. Our results showed that the highest effect of all amino acids was in *M. piperita* ‘Swiss’, while in *M. piperita* ‘Granada’, only in plants treated with tyrosine of 100 mg L^−1^ concentration was rutin increased. In *M. spicata* ‘Moroccan’ and in *M. spicata* ‘Crispa’ under foliar application with water, the positive correlation between the amounts of rutin and isoquercetin (r = 0.852, *p* < 0.05) was established. The results of our present study also indicate positive correlations between the amounts of rutin and quercetin (r = 0.679, *p* < 0.05), as well as between quercetin and total flavonoids (r = 0.822, *p* < 0.05). Isoquercetin is a glucosylated form of quercetin, known as a strong chemoprotectant against cardiovascular diseases, asthma, and diabetes [73,74,75,76]. The highest effect of amino acids on isoquercetin content was in *M. piperita* ‘Swiss’ plants. In *M. spicata* ‘Crispa’, a positive effect was found when phenylalanine and tryptophan of 200 mg L^−1^ concentration were applied to plants, while in *M. piperita* ‘Granada’, a positive effect was found with tyrosine of 100 mg L^−1^ concentration. Results showed that water application also had a positive effect on this compound content in some mint cultivars. Isoquercetin is formed from phenylalanine and tyrosine, with *p*-coumaroyl-Coa and Malonyl-Coa, from the reaction of chalcone synthase, other enzymes, and chemical reactions [77]. Myricetin synthesis starts from naringenin, flavonone precursor for all flavonols; the action of naringenin 3-dioxygenase produces dihydrokaempferol. Then, the flavonol synthase transforms this intermediate into kaempferol while the flavonoid 3′,5′-hydroxylase transforms kaempferol to myricetin [78]. The most effective foliar application on myricetin content was that of phenylalanine at 200 mg L^−1^ concentration and tyrosine at 100 mg L^−1^ concentration, depending on the variety of mint. The positive effect of phenylalanine on Myricetin 3-O-glucuronide content in Garnacha grapes was observed by Portu et al. [79]. According to Singh et al. [80], quercetin facilitates seed germination, pollen growth, antioxidant machinery, and photosynthesis, and is important for plant growth and development. The amount of quercetin was positively influenced in all mint varieties, but the greatest effect occurred in *M. piperita* ‘Swiss’, *M. piperita* ‘Granada’, and in *M. piperita* ‘Multimentha’ mint. Tryptophan of 200 mg L^−1^ concentration in *M. piperita* ‘Swiss’ increased quercetin by 2.31 times, while tyrosine of 100 mg L^−1^ concentration in *M. spicata* ‘Crispa’ increased it by 12.61 times. The effects of tryptophan may be related to the fact that tryptophan is a precursor of auxin. Lewis et al. [81] determined that, despite the effect of flavonoids on the inhibition of auxin, this hormone has an influence on the enzymes chalcone synthase, chalcone isomerase, and flavonone 3-hydroxylase, and on flavone synthase activity, which are involved in flavonoid synthesis. The effect of tyrosine and phenylanine at 200 mg L^−1^ concentration on the quercetin content in adventitious roots of *Gynura procumbens* and in cell cultures of *Citrullus colocynthis* (Linn.) were reported by (Meena et al. [82], Noviyanti et al. [83]). The amount of apigenin in mint varieties was both positively and negatively influenced, depending on amino acids, their concentration, and mint variety. Significantly, the highest amount of apigenin was observed in *M. piperita* ‘Swiss’ plants sprayed with water, triptophan, and 100 mg L^−1^ tyrosine solutions; this increased apigenin by 2.11, 2.51, and 2.80 times, respectively. Apigenin is synthesized from naringenin by chalcone isomerase and flavone synthase enzymes [84]. The biosynthesis of kaempferol starts with phenylalanine, which is converted into 4-coumaroyl-CoA. This enzyme combines with three molecules of malonyl-coA to form naringenin chalcone (tetrahydroxychalcone) through the action of the enzyme chalcone synthase [84,85]. The results showed that the highest amount of kaempferol was identified in *M. spicata* ‘Moroccan’ and in *M. piperita* ‘Swiss’ mint sprayed with tyrosine of 200 mg L^−1^ concentration (62.50 and 10.10 mg 100 g^−1^). In terms of efficiency, this amino acid increased kaempferol by 4.40 and 2.19 times. Phenylalanine of 100 and 200 mg L^−1^ concentration in *M. piperita* ‘Multimentha’ increased kaempferol by 2.55 and 11.04 times, respectively, while the tryptophan of 100 mg L^−1^ concentration produced the highest increase of kaempferol, by 6.89 times. Foliar application with amino acids showed a positive influence in *M. piperita* ‘Granada’ mint, except for tryptophan of 100 mg L^−1^ concentrated, which produced negative effects. According to scientific studies, amino acids have a positive effect on kaempferol synthesis in *Gynura procumbens* Lour., *Gardenija jasmioides* Sims., *Ginkgo biloba* L., and *Tylophora indica* Burm. F. plants [72,83,86,87,88]. As seen from the results, foliar application with aromatic amino acids allows for modulating essential oil content, essential oil odor profile, and flavonoid content and their chemical composition in mint, increasing the possibilities for application and the expansion of the scope of their use with this potentially economically important industrial plant.

## 4. Material and Methods

### 4.1. Experimental Sites and Soil

The research was conducted from 2017 to 2019 at Aleksandras Stulginskis University, the name of which, in 2019, was changed to Vytautas Magnus University Agriculture Academy. *M. spicata* ‘Moroccan’, *M. spicata* ‘Crispa’, *M. piperita* ‘Granada’, *M. piperita* ‘Swiss’, and *M. piperita* ‘Multimentha’ were planted on 4–5 May 2017; field location coordinates are 54°53′08, 9″ N, 23°50′08, 02″ E. The soil at the experimental site consists of ground moraine (ground glacial formations) covered with limnoglacial sediments. The soil at the experimental site was silty loam (46% sand, 42% silt, and 12% clay). Split-plot was used in the experimental design in four replications and the plot area was 800 m^2^ consisting of five rows (4 m in length and 50 cm between rows). The arable soil layer pH was between 7.20 and 7.70; mineral nitrogen was from 5.75 to 8.76; P_2_O_5_ was from 249 to 260; and K_2_O was from 141 to 148 mg kg^−1^ of soil. The mint plants were sprayed with aromatic amino acids L-phenylalanine, L-tryptophan, and L-tyrosine at two concentrations: 100 mg L^−1^ and 200 mg L^−1^, three times at a 15-day interval. The rate of applied solutions was 450 L ha^−1^. Tween 20 was added to the spraying solution as a surfactant. The first spraying was at the BBCH 21 stage of mint development. Foliar spray with water was used as a positive control. The effects of all treatments were compared with negative control (without spraying). The plants were harvested 15 days after the last spray at the BBCH 65 stage of mint development [89].

### 4.2. Meteorological Conditions

To determine the temporal variation of meteorological parameters and their impact on plant growth, the Sielianinov hydrothermal coefficient was used [90]. The month’s classification was carried out according to Skowera and Pula [91]: k ≤ 0.4 was extremely dry, 0.4 < k ≤ 0.7 was very dry, 0.7 < k ≤ 1.0 was dry, 1.0 < k ≤ 1.3 was quite dry, 1.3 < k ≤ 1.6 was optimal, 1.6 < k ≤ 2.0 was quite wet, 2.0 < k ≤ 2.5 was wet, 2.5 < k ≤ 3.0 was very wet, and k > 3.0 was extremely wet. April in 2017 was extremely wet before the planting of mint (K =15.1) (Table 8). May was a month of dry stress in all years (K = 0.3, k = 0.4, and K = 1.1). June in 2017 was quite wet, while in 2018 it was quite dry, and in 2019 it was dry. Only in July 2017 were conditions optimal for plant growth. August every year was quite dry (K = 1.0, K = 1.1, and K = 1.2) (Table 8). The meteorological conditions were very variable every year and plants grew under stress conditions.

### 4.3. Chemicals

Methanol, acetonitrile, and external standards, such as quercetin-3-O-rutinoside, quercetin-3-O-glucoside, quercetin, and kaempferol acid, with a purity of 99.5%, were purchased from Sigma-Aldrich and Fluka (Warsaw, Poland).

### 4.4. Methods of Sample Preparation

For analysis, a total of 100–200 g samples were taken from each replicate. Essential oil content was determined in dried substances, while flavonoid content and chemical composition were determined in lyophilized substances. Mint after harvest was dried in the drying oven (Termaks TS-8265, Bergen, Norway) at 30 °C for 24 h and lyophilized in the lyophilizator (SCANVAC Coolsafe 55-9, Denmark) at −60 °C for 24 h, and finally ground to a fine powder in a laboratory mill (Grindomix GM 200, Retsh GmbH, Haan, Germany). Prior to analyses of essential oil content, the samples were stored at room temperature, and flavonoids compounds were stored at −80 °C. Essential oil content and essential oil odor profile was determined in 2017–2019, and flavonoid content and chemical composition in 2017–2018.

### 4.5. Essential Oil Content

The amount of essential oil (%) was determined by extracting it from dried plant samples by hydrodistillation method using Clevenger-type apparatus, pouring 500 mL of water on 25 g of mint and boiling it for 4 h (LST EN ISO 6571). The amount of essential oil was calculated by Rubinskienė et al. [92]. The odor profile of mint essential oil was determined using E-nose Heracles II (Alpha M.O.S., Toulouse, France) which is based on ultra-fast gas chromatography. Analysis was conducted by using a cooled Tenax trap (Shimadzu, USA); two different polarity columns, non-polar MXT-5 (5% diphenyl) and semi-polar MXT-1701 (14% cyanopropylphenyl), of 10 m length and 180 µm diameter (Restek, Bellefonte, PA, USA); two flame detectors, FID1 and FID2 (Restek, USA); automatic sampler HS 100 (CTC Analytics AG, Zwingen, Switzerland); and hydrogen gas generator Alliance (Innovative Gas Systems Inc., USA). Briefly, 20 µL of mint essential oil was incubated at 40 °C temperature for 300 s under agitation (500 rpm), volume of the injection sample into the GC system was 1000 µL at a flow rate of 125 μLs^−1^. The carrier gas was hydrogen. For determination and sensory description AlphaSoft, V12 program, and Arochembase, V5 database (Alpha M.O.S., France) were used. Possible aromatic compounds are presented in Appendix A.

### 4.6. Flavonoid Content

Flavonoids were measured by HPLC method [92]. A total of 100 mg of freeze-dried mint powder was mixed with 5 mL of 80% methanol in a plastic test tube, and then mixed thoroughly by vortex and incubated in an ultrasonic bath for 15 min at 30 °C. The samples were centrifuged at 5000 rpm. Then, 1 mL of extract taken from the test tube was repeatedly re-centrifuged at 12,000 rpm. An amount of 500 µL of the extract was taken for HPLC vials and analyzed. A Synergy Fusion-RP 80i Phenomenex column (250 × 4.60 mm) was used for the analysis of flavonoids. Shimadzu equipment with two pumps (LC-20AD), a controller (CBM-20A), a column oven (SIL-20AC), and UV-vis spectrometer (SPD-20 AV) were used to carry out the analysis. The gradient flow was applied along with two mobile phases—10% (*v*/*v*) acetonitrile and ultrapure water (solvent A) and 55% (*v*/*v*) acetonitrile and ultrapure water solvent B, pH 3—the gradient program used was: 0-21 min, 95% solvent A and 5% solvent B; 22–25 min, 50% solvent A and 50% solvent B; 26–27 min, 20% solvent A and 80% solvent B; 28–32 min, 20% solvent A and 80% solvent B; 32–36 min, 95% solvent A and 5% solvent B. The analysis duration was 36 min, flow was 1 mL min^-1^, and wavelength for flavonoids was 360 nm. The phenolic compounds were identified by using 99.9% pure standards (Sigma-Aldrich, Warsaw, Poland) and the specified analysis times for the standards. The standard curves are presented in Appendix A.

### 4.7. Statistical Analysis

All analyses were performed in triplicate. The data analysis was conducted with STATISTICA version 12 software (StatSoft, Inc., Tulsa, OK, USA). The interaction between years and essential oil and flavonoid content was calculated using a two-way analysis of variance (ANOVA). Fisher’s test was applied to assess significant differences (*p* < 0.05) between samples. Then, the results were analyzed using a one-way analysis of variance (ANOVA). Tukey’s honestly significant difference test (HSD) was applied to assess significant differences between mean values (*p* < 0.05). The relationship between the values was determined using Pearson’s linear correlation coefficient (*p* < 0.05). Principal component analysis (PCA) was performed to evaluate the relationships between the applications of the amino acids, mint variety, and chemical content, as well as hierarchical cluster analysis (HCA), which was performed with XLSTAT software version 2019.2.2 (Addinsoft, Paris, France) to categorize the mint based on its flavonoid compound content. Aiming to assess the impact of amino acids on the aroma profile of mint, a principal component analysis (PCA) of essential oil was conducted, which was implemented by using the Alpha M.O.S. Heracles II device.

## 5. Conclusions

Secondary metabolites are responsible for many plant functions and play an important role in the human body. For these reasons, these biologically active compounds are important in many industrial fields. Therefore, promoting the synthesis of these compounds in mint, not only in vitro and in vivo, but in field conditions as well, is economically useful. Foliar spray with aromatic amino acids can increase the amount of essential oil, and total flavonoids, can change the odor profile of mint essential oil and the chemical composition of flavonoids. The influence of amino acids on the secondary metabolites depended on a particular compound whose synthesis is promoted, depending on the mint variety. The influence of amino acids on the essential oil content was only found in *M. piperita* ‘Granada’ plants when plants were sprayed with phenylalanine of 100 mg L^−1^ concentration, in which case the amount of this compound in these varieties of mint increased by 0.53 percentage units. Tyrosine of 100 mg L^−1^ concentration was the most effective in changing the essential oil odor profile in *M. spicata* ‘Crispa’ and *M. spicata* ‘Moroccan’ plants. The greatest amount of flavonoids, rutin, and quercetin were in *M. piperita* ‘Swiss’ plants, while in *M. spicata* ‘Moroccan’ and in *M. piperita* ‘Granada’ plants, amino acids showed the greatest effect on kaempferol. The largest amount of myricetin was identified in mint of the *M. piperita* ‘Granada’ species sprayed with phenylalanine solutions of 100 mg L^−1^ concentration. The greatest amount of apigenin was found in *M. spicata* ‘Crispa’ spearmint sprayed with phenylalanine solutions of 200 mg L^−1^ concentration. The results of the research create preconditions for purposeful improvement of technology applied when growing mint outdoors using amino acids and provide the opportunity to purposefully change the quantitative composition of secondary metabolites found in these plants. However, further research and investigations are essential for the modulation of the chemical composition of essential oil.

## Figures and Tables

**Figure 1 plants-11-02938-f001:**
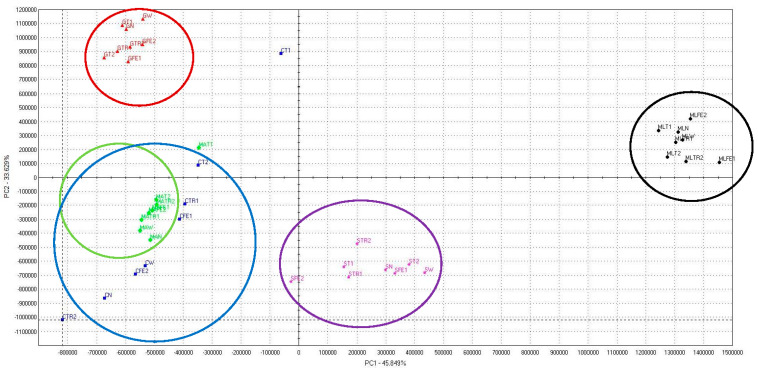
Principal component analysis (PCA) for volatile compounds of different mint cultivars sprayed with amino acids.

**Figure 2 plants-11-02938-f002:**
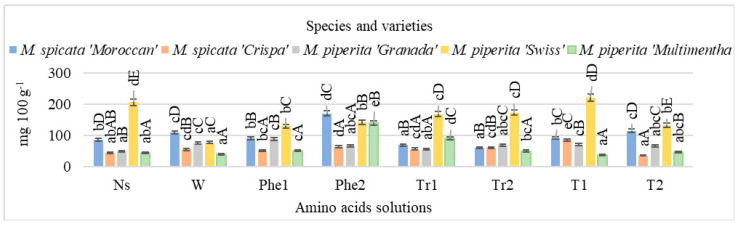
Total flavonoid content in mint influenced by foliar application of amino acids, mg 100 g^−1^ DM in 2017–2018 (NS: unsprayed; W: water; Phe1: phenylalanine, 100 mg L^−1^; Phe2: phenylalanine, 200 mg L^−1^; Tr1: tryptophane100 mg L^−1^; Tr2: tryptophane, 200 mg L^−1^; T1: tyrosine, 100 mg L^−1^; T2: tyrosine, 200 mg L^−1^; means marked with different uppercase letters (A, B, C…) indicate significant difference between varieties at *p* < 0.05; means marked with different lowercase letters (a, b, c…) indicate significant difference between foliar application with amino acids at *p* < 0.05).

**Figure 3 plants-11-02938-f003:**
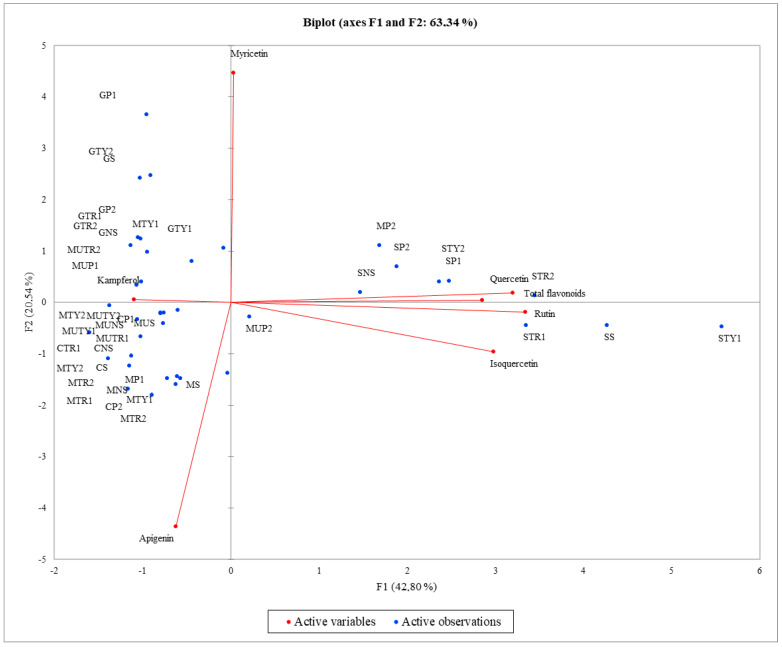
Principal component analysis (PCA) for total amount of flavonoid their chemical composition in different varieties of mint sprayed with different amino acids.

**Figure 4 plants-11-02938-f004:**
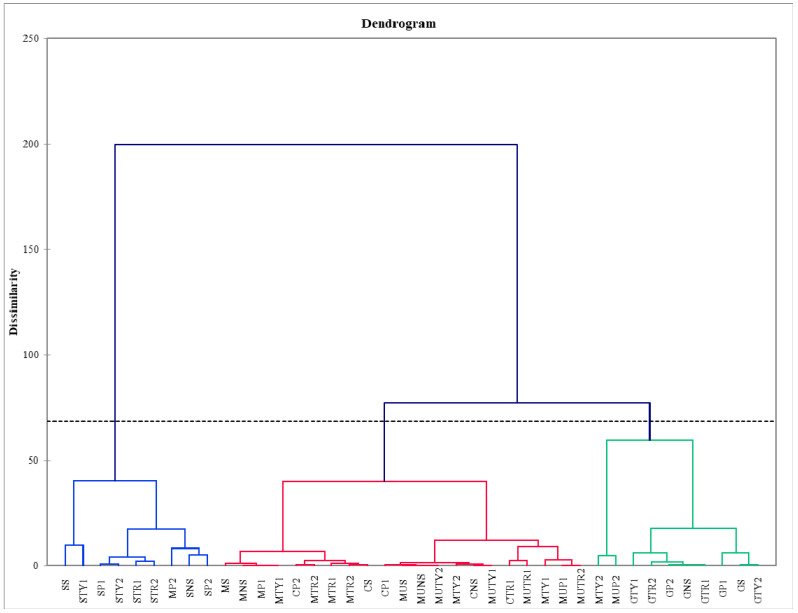
Hierarchical clustering analysis (HCA) of total amount of flavonoids in the chemical composition of different varieties of mint sprayed with different amino acids.

**Table 1 plants-11-02938-t001:** Essential oil content in mint influenced by foliar application of amino acids, mg 100 g^−1^ DM in 2017–2018.

Treatment	Species/Varieties				
	*M. spicata* ‘Moroccan’	*M. spicata* ‘Crispa’	*M. piperita* ‘Granada’	*M. piperita* ‘Swiss’	*M. piperita* ‘Multimentha’
Unsprayed	2.79 ± 0.05 abBC	1.59 ± 0.01cA	1.41 ± 0.30 aA	3.24 ± 0.71 bC	2.12 ± 0.11 aAB
Water	2.26 ± 0.15 aC	1.58 ± 0.01 cA	1.71 ± 0.03 abA	2.05 ± 0.03 aB	1.62 ± 0.03 aA
Phenylalanine 100 mg L^−1^	2.83 ± 0.03 abB	1.51 ± 0.05 cA	1.94 ± 0.15 bAB	2.85 ± 0.51abB	1.92 ± 0.67 aAB
Phenylalanine 200 mg L^−1^	3.09 ± 0.02 bD	1.20 ± 0.02 abA	1.50 ± 0.15 abAB	2.20 ± 0.24 abC	1.54 ± 0.03 aB
Tryptopan 100 mg L^−1^	2.75 ± 0.79 abC	1.25 ± 0.07 abA	1.44 ± 0.07 aAB	2.40 ± 0.64 abABC	2.54 ± 0.36 aBC
Tryptopan 200 mg L^−1^	3.09 ± 0.23 bC	1.23 ± 0.03 abA	1.68 ± 0.35 abAB	2.34 ± 0.13 abB	2.09 ± 0.44 aB
Tyrosine 100 mg L^−1^	2.71 ± 0.01abC	1.33 ± 0.10 bA	1.36 ± 0.05 aA	2.55 ± 0.20 abBC	2.06 ± 0.45 aB
Tyrosine 200 mg L^−1^	3.04 ± 0.16 abC	1.19 ± 0.01 aA	1.37 ± 0.03 aA	2.65 ± 0.20 abC	2.11 ± 0.30 aB

Mean ± standard deviations marked with different uppercase letters (A, B, C…) in the rows are significantly different at *p* < 0.05; means ± standard deviations marked with different lowercase letters (a, b, c…) in the columns are significantly different at *p* < 0.05.

**Table 2 plants-11-02938-t002:** Rutin content in mint influenced by foliar application of amino acids, mg 100 g^−1^ DM in 2017–2018.

Treatment	Species/Varieties				
	*M. spicata* ‘Moroccan’	*M. spicata* ‘Crispa’	*M. piperita* ‘Granada’	*M. piperita* ‘Swiss’	*M. piperita* ‘Multimentha’
Unsprayed	7.01 ± 0.52 aA	8.73 ± 1.73 bcAB	10.50 ± 0.75 cdABC	12.30 ± 3.11 aBC	13.04 ± 1.06 cC
Water	21.33 ± 3.13 bB	8.40 ± 0.52 bcA	9.08 ± 1.03 bcA	81.15 ± 7.49 dC	15.95 ± 1.12 cAB
Phenylalanine 100 mg L^−1^	7.79 ± 0.57 aA	10.32 ± 0.94 cA	5.77 ± 0.90 abA	35.86 ± 11.62 bB	12.28 ± 0.90 cA
Phenylalanine 200 mg L^−1^	8.46 ± 0.57 aAB	8.08 ± 0.59 bcAB	5.45 ± 0.51 aA	34.27 ± 3.17 bC	11.80 ± 1.47 cB
Tryptopan 100 mg L^−1^	8.50 ± 0.71 aBC	7.46 ± 0.69 bAB	12.20 ± 0.75 cdeC	54.28 ± 3.74cD	4.01 ± 0.32 aA
Tryptopan 200 mg L^−1^	6.25 ± 0.47 aA	8.72 ± 0.88bcAB	11.67 ± 0.77 cdB	41.09 ± 2.84 bcC	12.14 ± 3.61 cB
Tyrosine 100 mg L^−1^	7.79 ± 0.50 aAB	4.89 ± 0.30 aA	15.32 ± 2.99 eB	82.80 ± 6.53dC	7.72 ± 0.65 abAB
Tyrosine 200 mg L^−1^	6.91 ± 0.51 aA	4.25 ± 0.51 aA	13.41 ± 0.95 deB	47.57 ± 3.28 bcC	13.27 ± 0.89 cB

Mean ± standard deviations marked with different uppercase letters (A, B, C…) in the rows are significantly different at *p* < 0.05; means ± standard deviations marked with different lowercase letters (a, b, c…) in the columns are significantly different at *p* < 0.05.

**Table 3 plants-11-02938-t003:** Isoquercetin content in mint influenced by foliar application of amino acids, mg 100 g^−1^ DM in 2017–2018.

Treatment	Species/Varieties				
	*M. spicata* ‘Moroccan’	*M. spicata* ‘Crispa’	*M. piperita* ‘Granada’	*M. piperita* ‘Swiss’	*M. piperita* ‘Multimentha’
Unsprayed	3.03 ± 0.12 aA	3.48 ± 0.11 cdB	2.84 ± 0.16 abA	3.62 ± 0.13 aB	3.57 ± 0.09 cB
Water	2.86 ± 0.11 aA	3.41 ± 0.09 cB	2.96 ± 0.12 bA	7.30 ± 0.07 cD	3.94 ± 0.17 dC
Phenylalanine 100 mg L^−1^	3.05 ± 0.12 aAB	3.72 ± 0.09 dB	2.43 ± 0.16 aA	5.34 ± 0.57 bC	2.48 ± 0.13 aA
Phenylalanine 200 mg L^−1^	3.13 ± 0.10 aA	4.02 ± 0.10 eC	2.88 ± 0.11 bA	3.03 ± 0.13 aA	3.64 ± 0.17 cdB
Tryptopan 100 mg L^−1^	3.16 ± 0.12 aB	2.45 ± 0.11 aA	3.02 ± 0.12 bB	5.41 ± 0.29 bC	3.18 ± 0.16 bB
Tryptopan 200 mg L^−1^	2.95 ± 0.12 aB	4.60 ± 0.16 fC	3.03 ± 0.09 bB	5.14 ± 0.29 bD	2.33 ± 0.13 aA
Tyrosine 100 mg L^−1^	3.08 ± 0.13 aB	2.30 ± 0.13 aA	5.09 ± 0.30 cC	6.78 ± 0.25 cD	3.51 ± 0.11 bcB
Tyrosine 200 mg L^−1^	2.99 ± 0.12 aA	2.94 ± 0.09 bA	2.95 ± 0.12 bA	4.88 ± 0.10 bC	3.61 ± 0.11 cdB

Mean ± standard deviations marked with different uppercase letters (A, B, C…) in the rows are significantly different at *p* < 0.05; mean ± standard deviations marked with different lowercase letters (a, b, c…) in the columns are significantly different at *p* < 0.05.

**Table 4 plants-11-02938-t004:** Myricetin content in mint influenced by foliar application of amino acids, mg 100 g^−1^ DM in 2017–2018.

Treatment	Species/Varieties				
	*M. spicata* ‘Moroccan’	*M. spicata* ‘Crispa’	*M. piperita* ‘Granada’	*M. piperita* ‘Swiss’	*M. piperita* ‘Multimentha’
Unsprayed	5.85 ± 0.58 aA	6.38 ± 0.57 aA	13.31 ± 0.77 aB	5.81 ± 0.59 aA	5.69 ± 0.51 aA
Water	5.89 ± 0.60 aA	6.06 ± 0.58 aA	25.16 ± 0.86 cC	10.40 ± 0.70 cB	5.77 ± 0.58 aA
Phenylalanine 100 mg L^−1^	5.95 ± 0.58 aA	5.88 ± 0.57 aA	39.11 ± 1.43 dC	9.07 ± 0.67 bcB	5.91 ± 0.59 aA
Phenylalanine 200 mg L^−1^	17.52 ± 2.22 bB	6.25 ± 0.59 aA	15.46 ± 0.83 abB	8.28 ± 0.65 bA	7.12 ± 0.61 aA
Tryptopan 100 mg L^−1^	6.17 ± 0.57 aA	6.13 ± 0.61 aA	16.14 ± 0.67 abC	9.46 ± 0.67 bcB	6.11 ± 0.59 aA
Tryptopan 200 mg L^−1^	5.91 ± 0.55 aA	6.19 ± 0.59 aA	15.33 ± 0.83 abC	8.11 ± 0.55 bB	5.53 ± 0.57 aA
Tyrosine 100 mg L^−1^	6.07 ± 0.56 aA	8.92 ± 0.66 bB	16.96 ± 0.91 bD	12.59 ± 0.60 dC	6.18 ± 0.59 aA
Tyrosine 200 mg L^−1^	6.04 ± 0.57 aA	5.82 ± 0.60 aA	25.83 ± 1.83 cC	9.20 ± 0.58 bcB	6.13 ± 0.63aA

Mean ± standard deviations marked with different uppercase letters (A, B, C…) in the rows are significantly different at *p* < 0.05; means ± standard deviations marked with different lowercase letters (a, b, c…) in the columns are significantly different at *p* < 0.05.

**Table 5 plants-11-02938-t005:** Quercetin content in mint influenced by foliar application of amino acids, mg 100 g^−1^ DM in 2017–2018.

Treatment	Species/Varieties				
	*M. spicata* ‘Moroccan’	*M. spicata* ‘Crispa’	*M. piperita* ‘Granada’	*M. piperita* ‘Swiss’	*M. piperita* ‘Multimentha’
Unsprayed	29.58 ± 0.28 cB	4.15 ± 0.09 aA	2.76 ± 0.10 aA	45.53 ± 2.31 aC	2.98 ± 0.09 aA
Water	38.44 ± 1.51 dC	3.59 ± 0.09 aA	16.25 ± 0.24 dB	89.32 ± 1.97 eD	3.10 ± 0.13 aA
Phenylalanine 100 mg L^−1^	32.68 ± 0.86 cdD	9.11 ± 2.65 bA	17.52 ± 0.43 eC	69.29 ± 0.43 cE	14.24 ± 0.16 bB
Phenylalanine 200 mg L^−1^	115.91 ± 5.84 eE	3.60 ± 0.10 aA	17.63 ± 0.14 eB	87.67 ± 0.56 eD	55.50 ± 0.99 eC
Tryptopan 100 mg L^−1^	6.03 ± 0.13 aB	3.71 ± 0.10 aA	2.85 ± 0.13 aA	76.84 ± 0.45 dD	31.98 ± 1.67 dC
Tryptopan 200 mg L^−1^	13.32 ± 0.20 bB	3.47 ± 0.10 aA	3.44 ± 0.13 bcA	104.83 ± 1.52 gD	18.21 ± 0.34 cC
Tyrosine 100 mg L^−1^	33.55 ± 0.62 cdB	52.35 ± 0.32 cC	2.92 ± 0.11 abA	97.37 ± 4.35 fD	3.04 ± 0.13 aA
Tyrosine 200 mg L^−1^	14.03 ± 0.14 bC	5.83 ± 0.41 aB	3.98 ± 0.09 cAB	56.09 ± 1.65 bD	3.81 ± 0.17 aA

Mean ± standard deviations marked with different uppercase letters (A, B, C…) in the rows are significantly different at *p* < 0.05; means ± standard deviations marked with different lowercase letters (a, b, c…) in the columns are significantly different at *p* < 0.05.

**Table 6 plants-11-02938-t006:** Apigenin content in mint influenced by foliar application of amino acids, mg 100 g^−1^ DM in 2017–2018.

Treatment	Species/Varieties				
	*M. spicata* ‘Moroccan’	*M. spicata* ‘Crispa’	*M. piperita* ‘Granada’	*M. piperita* ‘Swiss’	*M. piperita* ‘Multimentha’
Unsprayed	26.58 ± 1.42 bcD	16.42 ± 1.36 bcC	6.35 ± 0.80 bA	5.81 ± 1.00 aA	10.03 ± 0.92 cdB
Water	25.53 ± 1.40 bcE	20.62 ± 1.57 cdD	3.59 ± 0.72 aA	12.24 ± 1.01 bC	8.62 ± 0.87 cB
Phenylalanine 100 mg L^−1^	26.26 ± 1.34 bcC	9.96 ± 2.19 abB	6.70 ± 0.79 bA	4.09 ± 0.72 aA	5.63 ± 0.76 abA
Phenylalanine 200 mg L^−1^	10.79 ± 0.88 aB	28.99 ± 1.50 eC	6.17 ± 0.76 bA	3.85 ± 0.72 aA	13.52 ± 2.84 deB
Tryptopan 100 mg L^−1^	28.87 ± 1.87 cC	17.05 ± 5.89 cB	6.04 ± 0.79 bA	14.60 ± 3.57 bB	14.19 ± 1.00 eB
Tryptopan 200 mg L^−1^	23.56 ± 1.54 bB	25.50 ± 1.53 deB	7.10 ± 0.82 bA	7.11 ± 1.04 aA	4.61 ± 0.75 aA
Tyrosine 100 mg L^−1^	26.84 ± 1.95 bcD	4.61 ± 0.75 aA	6.21 ± 0.80 bA	16.29 ± 1.68 bC	12.19 ± 1.00 deB
Tyrosine 200 mg L^−1^	22.58 ± 2.11 bC	9.50 ± 0.89 aB	3.65 ± 0.72 aA	4.95 ± 0.74 aA	10.31 ± 0.92 cdB

Mean ± standard deviations marked with different uppercase letters (A, B, C…) in the rows are significantly different at *p* < 0.05; means ± standard deviations marked with different lowercase letters (a, b, c…) in the columns are significantly different at *p* < 0.05.

**Table 7 plants-11-02938-t007:** Kaempferol content in mint influenced by foliar application of amino acids, mg 100 g^−1^ DM in 2017–2018.

Treatment	Species/Varieties				
	*M. spicata* ‘Moroccan’	*M. spicata* ‘Crispa’	*M. piperita* ‘Granada’	*M. piperita* ‘Swiss’	*M. piperita* ‘Multimentha’
Unsprayed	14.19 ± 0.80 abB	5.81 ± 0.37 aA	12.74 ± 0.73 aB	4.62 ± 0.54 aA	4.45 ± 0.56 aA
Water	15.58 ± 1.43 abB	13.10 ± 1.03 abB	18.22 ± 1.37 bC	6.77 ± 0.22 cA	6.06 ± 0.35 abA
Phenylalanine 100 mg L^−1^	15.39 ± 1.33 abCD	12.97 ± 1.00 abBC	17.44 ± 1.27 bD	5.77 ± 0.38 bA	11.33 ± 0.40 bB
Phenylalanine 200 mg L^−1^	14.83 ± 0.84 abBC	12.55 ± 0.57 abAB	19.50 ± 1.55 bcC	5.76 ± 0.32 bA	49.14 ± 5.74 dD
Tryptopan 100 mg L^−1^	17.04 ± 0.98 abAB	20.37 ± 14.37 bAB	16.36 ± 0.86 abAB	8.07 ± 0.13 dA	30.66 ± 3.01 cB
Tryptopan 200 mg L^−1^	8.80 ± 0.08 aA	12.37 ± 0.49 abB	28.56 ± 2.87 dC	7.12 ± 0.36 cA	7.39 ± 0.15 abA
Tyrosine 100 mg L^−1^	14.79 ± 0.81abB	12.43 ± 0.56 abB	23.56 ± 2.59 cC	5.31 ± 0.45 abA	5.28 ± 0.45 abA
Tyrosine 200 mg L^−1^	62.50 ± 7.09 cC	7.64 ± 0.29 abA	17.51 ± 1.29 bB	10.10 ± 0.23 eAB	8.53 ± 0.11 abA

Mean ± standard deviations marked with different uppercase letters (A, B, C…) in the rows are significantly different at *p* < 0.05; mean ± standard deviations marked with different lowercase letters (a, b, c…) in the columns are significantly different at *p* < 0.05.

**Table 8 plants-11-02938-t008:** Sielianinov coefficient during vegetation period of mint in 2017–-2019.

Year	Month	Hydrotermal Sielianinov Coefficient	Classification by the Month, by Skowera, Pula (2004)
2017	April	15.1	Extremely wet
	May	0.3	Extremely dry
	June	1.6	Quite wet
	July	1.5	Optimal
	August	1.0	Quite dry
2018	April	2.9	Very wet
	May	0.4	Extremely dry
	June	1.1	Quite dry
	July	2.2	Wet
	August	1.1	Quite dry
2019	April	0.0	Extremely dry
	May	1.1	Quite dry
	June	0.8	Dry
	July	1.1	Quite dry
	August	1.2	Quite dry

## Data Availability

The data were taken from doctoral thesis of A. Velička, “The impact of genotype and aromatic amino acids on the formation and chemical composition of the biological potential of mint (*Mentha* L.)”. Results were not published in another scientific journal.

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
