# Peer review of "Impact of Foliar Application of Amino Acids on Essential Oil Content, Odor Profile, and Flavonoid Content of Different Mint Varieties in Field Conditions"

_plants, 2022, doi:10.3390/plants11212938_

Round 1

Reviewer 1 Report

Figure 1. – check the writing type

Figure 2. Essential oil Principal component analysis (PCA) of different mint cultivars sprayed with amino ac-ids – I recommend a better resolution

Figure 3. - check the writing type

Bibliography - I recommend a revision are several types of writing

I recommend a review of the way of drafting and regarding the tables found in the article

The quercetin content in mints fluctuated from 2.76 in M. piperita 'Granada' to 115.91 mg 100 g-1 in M. spicata 'Moroccan'. - unit of measurement

Instead of Figure 1, I recommend a table that is much more suggestive

I recommend a more thorough revision regarding the way of drafting and writing the article

Author Response

Thank you very much for your sugestions how o improve article.

Reviewer 2 Report

The manuscript represents an interesting study about the effect of foliar application of amino acids on the quality of different species of mint. However, the manuscript does not have a number of lines, which makes it difficult to review.

The identification and quantification of phenolic compounds presents several errors, which is essential for the validity of the reported results. On the other hand, the manuscript presents deficiencies in its format, figures, writing and mainly in the description of the results, which should be improved.

11.       In the abstract, in mg L-1, -1 should be superscripted. Also, the scientific names should be in italics.

22.       In the introduction, the information regarding the detected secondary metabolites and the biological effects reported is very descriptive. It is necessary to add quantitative information in addition to more references. It is unthinkable that this high number of effects has been studied in only two studies.

33.       On the other hand, it is necessary to add information about the absorption and transport mechanism of aminoacids in the plant.

44.       It is necessary to add a hypothesis at the end of the introduction.

55.    In the results, the description of the essential oil content is very scarce and must be highly improved. Information about the effect of aminoacids should be expanded. On the other hand, the values ​​indicated just before figure 1 (1.38 times) do not correspond to what is reported in the figure.

66.       The quality of figures 1 and 3 must be improved. Please, remove dividing lines and decimals on the Y axis. Statistical information should go over error bars and numbers removed for better understanding. In both figures and in all the tables, the control information (Ns) must be included first and then W and the other treatments.

77.       It is indicated that 51 volatile compounds were detected, however, the qualitative and quantitative information is not reported. This information must be included in the results and in the discussion using multivariate analysis together with phenolics.

88.       Figure 2 should be enhanced, ideally using different colors per treatment or species, for better understanding. On the other hand, the description of these results must be rewritten, since it is not understood.

99.       There are several errors in the described results, but it is difficult to detail as we do not have the line numbers. For example, in M. spicata in figure 3. Phenylalanine in table 2, M. piperita in table 2. M piperita in Table 4. All text must be revised.

110.   It is necessary to incorporate a PCA with all the studied variables, including the components of the essential oil and phenolic compounds, in order to discriminate whether the profiles of these compounds allow the different species of mint to be discriminated.

111.   Lines 85-87: several opinions are mentioned, these must be discussed.

112.   Lines 97-99: the increase is in number of compounds? Concentrations?

113.   Line 163: glcU?

114.   In the methodology, ml should be replaced by mL and ul by uL, according to the international measurement system.

115.   In point 4.6 it is necessary to provide information about the identification of the phenolic compounds (comparison with retention time of standards? Mass spectrometry?). It is strange that the detected compounds corresponds to aglycones (excepting rutin), when in this type of structures glycosidated derivatives are predominant.

116.   Why the quantification of phenolic was carried out at 250 nm? maximum absorption for all these compounds is close to 360 nm? 250 nm gives very low sensitivity for this determination. What method and standards were used to quantify?

117.   The reviewed literature needs to be updated. Only 23 of 86 references correspond to the last 5 years.

Author Response

Thank you very much for detail the review and for your recommendation and suggestions on how to improve our manuscript and increase its quality to the requirements of the “Plants” journal

Reviewer 3 Report

The manuscript entitled "Impact of Foliar Application of Amino Acids on Essential Oil Content, Odor profile, and Flavonoids Content of Different Mints Varieties Under the Field Condition" addresses a suitable topic for the Plants Journal.

Secondary metabolites are responsible for many plant functions as well as they play an important role in the human body, and for this reason, these biologically active compounds are important in many industrial fields. So promoting the synthesis of these compounds in mints not only in vitro and in vivo but also under field conditions is economically useful. Foliar spray with aromatic amino acids can increase the amount of essential oil, and total flavonoids can change the odor profile of mint essential oil, and the chemical composition of flavonoids. The influence of amino acids on the secondary metabolites depends on a particular compound whose synthesis is promoted depending on the mint variety. The influence of amino acids on the essential oil content was only in M. piperita 'Granada' plants, while amino acids the most effective in the change of essential oil odor profile was in M. spicata 'Crispa' plants.

The manuscript is very well written and organized. The subject addressed is one of interest in the field of Plants.

This manuscript can be published after correcting the requirements of the Plants Journal regarding citations, Templeate, English language corrections and after improving the conclusion, which should provide an overview of the results derived from this manuscript.

Author Response

Thank you very much for the review and for your recommendation and suggestions.

Round 2

Reviewer 1 Report

Notice an improvement in terms of the article, and I congratulate you.

Author Response

Thank you very much.

Reviewer 2 Report

Thank you very much for your responses. The manuscript has been improved. However, there are some aspects that must be considered:

1.       In abstract, scientific names must be full written. For example, Mentha piperita.

2.       In the introduction, please replace:

….phenylalanine on 250 and 500 mg L-1 concentration increase compared with unsprayed plants amount of essential oil in Salvia officinalis plants 0.06-0.12 (ml 100 g−1 dw) and amount of 1,8 Cineole (0,97 %) and cam-phor (0.64 %)….

by

… phenylalanine on 250 and 500 mg L-1 concentration increase compared with unsprayed plants amount of essential oil in Salvia officinalis plants 0.06-0.12 (mL 100 g−1 DW) and amount of 1.8 Cineole (0.97 %) and cam-phor (0.64 %)…

Also, please define DW.

3.       Hypothesis must be rewritten and clarified.

4.       Comment 112: “Lines 97-99: the increase is in number of compounds? Concentrations?: Please add quantitative information (values).

5.       Figure 1S must be corrected. Information in X and Y axis, and also the used standards are missing. On the other hand, a calibration curve must be initialized from (0.0). In this case, also analytical parameters as detection and quantification limit and lineal range must be incorporated.

6.       If phenolic compounds were identified only by comparison of retention time with commercial standards, HPLC chromatograms of standards and samples and also UV-vis spectra, must be added in supplementary material.

Author Response

Thank you very much for your recomendations. 

Round 3

Reviewer 2 Report

The manuscript has been highly improved. However, an important point must be revised, because could imply several errors in the results

  1. In Figure S1: Please revise that in all calibration curves must be graphed the concentration in X axis   and the units of area in Y axis. However, in the manuscript was represented the contrary. Please check that the concentrations and analytical parameters are well calculated.

Also, in all graphs, please replace ug ml-1 by ug mL-1

Please, replace mirycetin by myricetin.

  1. In table S1: Please, replace:

-       Kempferol by kaempferol

-       Mirycytin by Myricetin

-       Add concentration units of LOD, LOQ and linear range.

Author Response

Thank you very much for your suggestions how to improve quality of article.

Round 4

Reviewer 2 Report

Dear authors. Thank you very much for your responses. The suplemmentary material has been improved.

Just some comments:

1.  In Figure S1 and Table S1, the units are ug or ug mL-1?. Please confirm.

2. In Table S1, please define LOD and LOQ.

3. In table S1, all linear ranges are incorrect. Linear range is defined as the range of concentrations between LOQ and limit of linearity. For this reason, the linear range corresponds to a range not an only value.

Author Response

Thank you very much once.
